# Mitochondrial and Redox Changes in Periodontitis and Type 2 Diabetes Human Blood Mononuclear Cells

**DOI:** 10.3390/antiox12020226

**Published:** 2023-01-18

**Authors:** Ildete L. Ferreira, Solange Costa, Bruno J. Moraes, Ana Costa, Olga Fokt, Daniela Marinho, Vera Alves, Isabel P. Baptista, A. Cristina Rego

**Affiliations:** 1CNC—Center for Neuroscience and Cell Biology, University of Coimbra, 3004-504 Coimbra, Portugal; 2IIIUC—Institute for Interdisciplinary Research, University of Coimbra, 3030-789 Coimbra, Portugal; 3Institute of Periodontology, Dentistry Department, Faculty of Medicine, University of Coimbra, Avenida Bissaya Barreto, 3000-075 Coimbra, Portugal; 4Institute of Immunology, Faculty of Medicine, University of Coimbra, 3004-504 Coimbra, Portugal; 5Coimbra Institute for Clinical and Biomedical Research (iCBR), Faculty of Medicine, University of Coimbra, 3000-548 Coimbra, Portugal; 6Institute of Biochemistry, Faculty of Medicine, University of Coimbra, Rua Larga, 3004-504 Coimbra, Portugal

**Keywords:** periodontitis, type 2 diabetes mellitus, oxidative stress, mitochondria, glutathione, peripheral blood mononuclear cells

## Abstract

Periodontitis (PDT) and type 2 diabetes (T2D) have demonstrated a bidirectional relationship and imbalanced oxidative stress linked to mitochondrial dysfunction. Therefore, we investigated mitochondrial and redox (de)regulation in peripheral blood mononuclear cells (PBMCs) in comorbid T2D-PDT, compared to PDT, T2D patients, and control individuals. PBMCs were analyzed for mitochondrial respiration, reactive oxygen species, antioxidant proteins, and expression of Nrf2-target genes. PDT and T2D-PDT patients exhibited altered periodontal clinical markers, while T2D and T2D-PDT patients displayed increased blood HbA1c. Decreased oxygen consumption and ATP production were observed in the PDT patient’s PBMCs. PDT and T2D-PDT PBMCs also evidenced increased H_2_O_2_ levels and reduced catalase levels (also detected in T2D patients), whereas a compromised glutathione cycle was observed in T2D-PDT patients. PBMCs from both T2D or T2D-PDT patients showed increased Nrf2 protein levels, enhanced GCL activity and GCL-catalytic subunit protein levels, and maintained GCLc, GST, and HO-1 mRNA levels. In contrast, the expressions of Nrf2-target genes were significantly diminished in the PDT patient’s PBMCs. Decreased SOD1 and GST mRNA levels were also observed in CD3+CD8+-lymphocytes derived from PDT and T2D-PDT patients. In conclusion, PBMCs from T2D-PDT patients showed major redox changes, while mononuclear cells from PDT patients showed mitochondrial deregulation and reduced expression of Nrf2-target genes.

## 1. Introduction

Periodontitis (PDT) is a chronic oral inflammatory disease, linked to an inappropriate host immune response to a dysbiotic microbiome, leading to bone resorption and tooth loss [1]. Major extrinsic risk factors favoring PDT development include smoking, obesity, and poor nutrition [2]. The estimated prevalence of severe periodontitis was around 14% in 2019 [3,4], being considered the 6th most prevalent disease worldwide [5].

Type 2 diabetes (T2D) is a metabolic disorder characterized by hyperglycemia associated with insulin resistance and a progressive loss of β-cell insulin secretion, being obesity, caused by a high caloric intake in combination with a sedentary lifestyle, a risk factor for T2D [6]. T2D has been implicated in a bi-directional relationship with PDT [7,8]; indeed, T2D patients exhibit an increased risk of PDT, contributing to systemic inflammatory burden and a decline in glycemic control [9], while individuals with PDT display poor glycemic control and increased risk of developing T2D [10]. In addition, hyperglycemia may lead to oxidative stress via several inflammatory-response pathways [11], but their role in periodontal pathogenesis remains elusive. Resistin, a proinflammatory adipokine related to insulin resistance and obesity [12], is also involved in PDT [13].

Mitochondria play a major role in generating cellular energy in the form of adenosine triphosphate (ATP) via oxidative phosphorylation (OXPHOS), being key in the regulation of intracellular calcium, redox signaling, heme synthesis, apoptosis, and inflammatory response [14]. T2D has been associated with mitochondrial dysfunction linked to oxidative stress [15]. Interestingly, diabetes induction along with PDT-like phenomena in rats caused severe mitochondrial dysfunction, with compromised ATP production, decreased mitochondrial DNA copy numbers, reduced gene expression of electron transport chain complex I subunits, and impaired mitochondrial biogenesis and oxidative stress in gingival samples, suggesting mitochondria as a therapeutic target for comorbid conditions [16]. Additionally, oxidative stress and nuclear factor erythroid 2-related factor 2 (Nrf2) downregulation may participate in the aggravation of periodontitis by diabetes in rats [17]. Recent findings have noted the importance of bioenergetic analysis through the measurement of oxygen consumption rates (OCR) in studying mitochondrial (dys)function in human immune cells derived from inflammatory diseases, such as T2D, lymphocytes, or the main population in isolated peripheral blood mononuclear cells (PBMCs) [18]. Moreover, current research suggests that reactive oxygen species (ROS) play a role in diabetic and periodontal complications, contributing to the impairment of antioxidants responsible for decreasing ROS levels and maintaining vascular health [19]. In light of these findings, a new research field is emerging bridging mitochondrial and redox deregulation in the study of mechanisms related to T2D and PDT, as well as in finding specific and sensitive biomarker(s) that can be used for risk and screening assessment of both pathologies.

Assuming that PBMCs reproduce systemic changes occurring in PDT and T2D, we hypothesized the synergic cytopathological effects of these two diseases associated with mitochondrial deregulation and redox modifications. Thus, the aim of this study was to investigate changes in mitochondrial bioenergetics and redox regulation in PBMCs from comorbid T2D-PDT patients, when compared to PDT or T2D patients and healthy controls. Using PBMCs, we analyzed mitochondrial functions and bioenergetics and altered redox status linked to hydrogen peroxide (H_2_O_2_) levels and antioxidant activity by focusing on the glutathione redox cycle, catalase, and superoxide dismutase-2 (SOD2) as antioxidant markers, the transcription factor nuclear factor erythroid 2-related factor 2 (Nrf2), and the expression of its target genes involved in the detoxifying and antioxidant responses, namely superoxide dismutase 1 (*SOD1*), heme oxygenase 1 (*HO-1*), glutathione S-transferase (*GST*), glutamate-cysteine ligase catalytic subunit (*GCLc*), and glutamate-cysteine ligase modifier subunit (*GCLm*).

## 2. Materials and Methods

### 2.1. Study Design

The present study follows the STROBE guidelines for reporting cohort and cross-sectional studies. Individuals attending a follow-up appointment at the clinic of periodontology of the Department of Dentistry of “Centro Hospitalar e Universitário de Coimbra” (CHUC) in Coimbra, Portugal, were recruited for this study. Individuals aged 39 to 61 years, presenting a minimum of 15 teeth and previously diagnosed with PDT, T2D, or both PDT and T2D, were probed for periodontitis-related clinical parameters. Similarly, an age-matched control group was randomly selected from non-diabetic and non-periodontal patients. Patients were categorized into three study groups: PDT: 10 subjects with a history of periodontitis and probing depths (PD) > 3 mm; T2D: 10 subjects with no history of periodontal disease, but with a previous diagnosis of T2D; T2D-PDT: 10 subjects with a history of periodontitis, with PD > 3 mm and previous diagnosis of T2D; and a Control group: 10 subjects with no history of either periodontal disease or T2D. Clinical parameters for PDT, namely PD, clinical attachment loss (CAL), and bleeding on probing (BOP), were assessed for all subjects using a periodontal probe (PCPUNC15; Hu-Friedy, Chicago, IL, USA). According to the multi-dimensional staging and grading system, periodontitis was characterized if interdental CAL was detectable at ≥2 non-adjacent teeth, probing depth over 3 mm, and bleeding on probing [20]. Supragingival plaque deposition was quantified at 6 sites per tooth, using a plaque disclosure solution (Dento Plaque; Pierre Fabre—Oral Care, Gien, FR), and presented as a percentage (plaque index—PI). All patients were evaluated for weight and height, and body mass index (BMI) was calculated for each individual according to the following formula: BMI = [weight (kg)]/[height (m)^2^], which corresponds to the body fat percentage, allowing us to classify if someone is within the interval of normal weight (18.5–24.9 kg/m^2^). Medications taken by the subjects were retrieved from individual questionaries filled out before the experiments started, and can be found summarized in Appendix A. Subjects’ rights were protected by the Ethical Board of the Faculty of Medicine, University of Coimbra, and written informed consent was granted by all subjects. About 15 mL of venous peripheral blood was collected by venipuncture into tubes containing ethylenediamine tetraacetic acid (EDTA) as an anticoagulant (Sarstedt, Newton, NC, USA).

### 2.2. Exclusion Criteria

Exclusion criteria included pregnancy/lactation, HIV or hepatitis (B, C), uncontrolled systemic diseases (except T2D) or neoplasms, medical conditions that prevented an oral exam, chronic antibiotic use or requiring antibiotic coverage for dental procedures, underwent therapy with corticosteroids and/or immunosuppressive treatment in the 3 months prior to the periodontal evaluation, chronic use of non-steroidal anti-inflammatory drugs, and long-term treatment with medications known to affect periodontal status (phenytoin, cyclosporine).

### 2.3. Glycated Hemoglobin Determination

Blood HbA1c levels were determined by high-performance liquid chromatography (HPLC; Bio-Rad, Hercules, CA, USA) at Clinical Pathology service, CHUC.

### 2.4. Resistin Levels

Resistin plasma levels were determined from a standard curve for resistin by enzyme-linked immunosorbent assay (ELISA kit; RayBiotech, Peachtree Corners, GA USA) following the fluorescence intensity produced at 450 nm, according to the manufacturer’s instructions. All samples were run in duplicates and resistin concentration was calculated from a standard curve for resistin.

### 2.5. Isolation of Peripheral Blood Mononuclear Cells

Blood samples were diluted in phosphate-buffered saline (PBS) containing (in mM): 137 NaCl, 2.7 KCl, 1.8 KH_2_PO_4_, 10 Na_2_HPO_4_·2H_2_O, pH 7.4, carefully layered onto Ficoll-Paque (GE Healthcare Bio-Sciences, Chicago, IL, USA) solution and then centrifuged at 2500 rpm for 20 min at 18 °C in a swing-out rotor, without brake. The PBMCs-containing ring was removed, further diluted with PBS, and centrifuged at 1500 rpm for 10 min at 18 °C. Cells were immediately used for functional studies or fluorescence-activated cell sorting (FACS), kept on NZYol for RNA extraction at −20 °C, or frozen as pellets at −80 °C.

### 2.6. Mitochondrial Oxygen Consumption Rate

Oxygen consumption rate (OCR) measurements in PBMCs were performed using the Seahorse extracellular flux analyzer (Agilent Technologies, Santa Clara, CA, USA) [21]. Quantification of basal and maximal respiration, ATP production, reserve capacity, H^+^ leak, non-mitochondrial respiration, and coupling efficiency was obtained, expressed in nmoles O_2_/mL/min/0.2 × 10^6^ cells. Bioenergetic health index (BHI), an alias for “mitochondrial health”, was calculated accordingly to the formula: BHI = log [(reserve capacity × ATP-linked)/non-mitochondria respiration × proton leak)] [22]. A detailed description of the procedures can be found in ‘Appendix A’.

### 2.7. Analysis of H_2_O_2_ Production in PBMCs

H_2_O_2_ production by PBMCs was evaluated using the Amplex Red assay. Briefly, 0.5 × 10^6^ cells were resuspended in Na^+^ medium containing (in mM): 140 NaCl, 5 KCl, 1 CaCl_2_, 1 MgCl_2_, 10 glucose, 10 Hepes, pH 7.4 plus Amplex Red reagent (10 μM; Molecular Probes. Eugene, OR, USA) and horseradish peroxidase (0.5 units per mL; Sigma-Aldrich, St. Louis, MO, EUA). The reaction of Amplex Red (10-acetyl-3.7-dihydroxyphenoxazin) in the presence of peroxidase, at 1:1 stoichiometry, produces resorufin, a red-fluorescent oxidation product. Fluorescence was followed at 37 °C for 20 min, at an excitation wavelength of 550 nm and an emission wavelength of 580 nm, using a Microplate Spectrofluorometer Gemini EM (Molecular Devices, San Jose, CA, USA). Results were expressed as RFU/minute.

### 2.8. Preparation of PBMCs’ Total Extracts

The PBMCs pellets were resuspended in 20 mM hypotonic potassium phosphate buffer, pH 7.5, snap-frozen three times in liquid nitrogen, and then stored at −80 °C [23]. Protein quantification was performed using the Bio-Rad protein assay (Bio-Rad, Hercules, CA, USA). PBMCs total extracts were further used for enzymatic activities or denatured for SDS-PAGE and Western blotting.

### 2.9. Evaluation of Enzyme’s Activity

Glutamate–cysteine ligase (GCL) and glutathione levels were measured by a fluorometric assay whereas glutathione reductase (GRed) and glutathione peroxidase (GPx) activities were determined by spectrophotometry, as detailed in the ‘Appendix A’.

### 2.10. Sample Preparation and Western Blotting

PBMCs total extracts were denatured with sample buffer containing 300 mM Tris-HCl pH 6.8, 12% SDS, 30% glycerol, 600 mM DTT, 0.06% bromophenol blue at 95 °C for 5 min. Equivalent amounts of protein (20 µg) were separated in 7.5–10% SDS-PAGE gel electrophoresis and electroblotted onto polyvinylidene difluoride (PVDF) membranes. The membranes were further blocked with 5% milk in TBS-Tween: 25 mM Tris-HCl, 150 mM NaCl, pH 7.6)/0.1% Tween-20 (*v*/*v*) for 1 h, at room temperature, and further incubated with primary antibodies against acetyl-SOD2-K68 (ab137037), SOD2 (ab13533), GCLc (ab17926), glutathione peroxidase 1 (GPx-1; ab108427), glutathione reductase (GRed; ab16801), catalase (ab15834), Nrf2 (ab31163) purchased from Abcam, Cambridge, UK and GAPDH (MAB374; Sigma-Aldrich, St. Louis, MO, EUA), prepared with 5% milk in TBS-Tween overnight at 4 °C with gentle agitation. Membranes were then incubated with anti-mouse (no. 1316) or anti-rabbit (no. 1317) IgG secondary antibodies (GE Healthcare Bio-Sciences, Chicago, IL, USA) prepared with 1% milk in TBS-Tween for 1 h, at room temperature. The amount of protein per lane was normalized to GAPDH (loading control). Immunoreactive bands were visualized by alkaline phosphatase activity after incubation with ECF reagent (GE Healthcare Bio-Sciences, Chicago, IL, USA) on BioRad ChemiDoc Touch Imaging System and quantified using Image Lab analysis software (Bio-Rad, Hercules, CA, USA).

### 2.11. Fluorescence-Activated Cell Sorting (FACS)

Cell sorting was performed to separate CD3+CD4+ and CD3+CD8+ lymphocyte populations. The absolute numbers of the PBMCs were determined using DXH 500 (Beckman Coulter, CA, USA) equipment in order to adjust the volume of each antibody. FITC mouse anti-human CD3, PE mouse anti-human CD4, and PerCP-Cy 5.5 mouse anti-human CD8 antibodies (BD Pharmingen, CA, USA) were used according to the manufacturer’s instructions and incubated for 15 min at room temperature, in the dark. Cells were further washed in PBS, centrifuged at 450× *g* for 5 min, and pelleted cells resuspended in PBS until sorting using a FACSAria III^TM^ from Becton Dickinson, CA, USA. About 10,000 events were acquired to define the gating strategy using morphologic and fluorescence parameters. Using a dot-plot from FSC (forward light scatter) vs. SSC (Side scatter), lymphocytes were gated and, after excluding doublet, lymphocytes subsets were identified by gating CD3+CD4+ and CD3+CD8+, and non-CD3+. Monocytes were initially selected by morphologic parameters, doublets were excluded, and monocytes were identified by CD4 positivity since we did not use a specific monocyte marker. No major differences were observed in the percentage of lymphocyte subpopulations and monocytes obtained from the different groups (Appendix A).

### 2.12. Analysis of Nrf2-Target Genes

Nrf2-target genes were evaluated by reverse transcription quantitative real-time PCR (RT-qPCR). Total RNA was extracted from total PBMCs, CD3+CD4+ and CD3+CD8+ sorted lymphocytes using NZYol (NZYTech Lisbon, Portugal; 0.8 mL/90–300 × 10^3^ cells), according to the instructions of the supplier. Briefly, cell lysates were incubated for 5 min at room temperature and chloroform (200 μL per 1 mL NZYol used) was added and mixed vigorously before incubating for another 2 min. After centrifugation at 12,000× *g* for 15 min at 4 °C, using an Eppendorf Centrifuge 5417R, the clear upper aqueous layer was transferred to a new tube, isopropanol (500 μL per 1 mL NZYol) was added and the samples incubated for 10 min at −20 °C and further centrifuged at 12,000× *g* for 10 min, at 4 °C. The RNA precipitate was washed with 75% ethanol and the samples were centrifugated at 12,000× *g* for 10 min, at 4 °C. The pellet was dried at room temperature and then solubilized in 20 μL of DEPC-treated water. RNA concentration was determined with NanoDrop 2000c spectrophotometer (Thermo Scientific, Waltham, MA, USA). Complementary DNA (cDNA) was then synthesized from 500 ng of total extracted RNA using the NZY First-Strand cDNA Synthesis Kit (NZYTech, Lisbon, Portugal), following the manufacturer’s instructions. PCR reactions were performed in 10 μL volumes containing 5 μL of iQ SYBR Green Supermix (BioRad, Hercules, USA), 300 nM of each primer (as described below), and 50 ng of cDNA template in a Bio Rad CFX96 Real-Time PCR Detection System using the following cycling conditions: initial denaturation at 95 °C for 3 min, followed by 40 cycles of denaturation at 95 °C for 15 s and annealing at 55–61.7 °C for 45 s. In the end, samples were subjected to a melting curve analysis in order to confirm the absence of unspecific amplification products and primers dimers. Samples containing no template were included as negative controls in all experiments. Reactions were run in duplicates and the analysis of gene expression was performed using the ΔΔCT method. Glyceraldehyde-3-phosphate dehydrogenase (GAPDH) was used as an internal control for all samples. PCR primer sequences used were as follows (forward primer/reverse primer):*SOD1* F: GGTGGGCCAAAGGATGAAGAG/R:CCACAAGCCAAACGACTTCC*HO-1* F: CCTGAGTTTCAAGTATCC/R:AACAACAGAACACAACAA*GST* F: GGAGGCAAGACCTTCATT/R:ATGGATCAGCAGCAAGTC*GCLc* F:GGCACAAGGACGTTCTCAAGT/R:CAAGACAGGACCAACCGGAC*GCLm* F:AACTCTTCATCATCAACTA/R:AACTCCATCTTCAATAGG*GAPDH* F: ATTCCACCCATGGCAAATTC/R:GGGATTTCCATTGATGACAAGC

### 2.13. Power and Sample Size

The present study presents a factorial design with two factors and two levels within each (diabetes: Y/N; periodontitis: Y/N). For sample size determination, the authors agreed that it would be clinically relevant if 20% of the variance in a normally distributed primary outcome variable would be attributable to the main effect “presence of periodontitis”. This would correspond to an effect size f of 0.5 (moderate to large). Based on these assumptions, we would require a total sample of 34 patients to detect this difference with 80% power at the (two-sided) 5% level. Assuming the effect size to be the same for both the presence of periodontitis and the presence of diabetes, the aforementioned 34 patients would provide sufficient power to detect the main effects and an interaction effect that is twice as large as the assumed main intervention effect. In order to provide an equal number of patients in each group and to compensate for possible sample collection/handling errors, the required number of patients was increased to 40, i.e., 10 patients per group. All calculations were performed using the “F tests family” and the “ANOVA: Fixed effects, special, main effects and interaction” function of G*Power Version 3.1.9.6.

### 2.14. Statistical Analysis

Data were analyzed using Microsoft Excel and GraphPad Prism 8 software (GraphPad, San Diego, CA, USA), and the results were expressed as the mean ± standard error of the mean (SEM). Sample normality was tested using the Shapiro–Wilk test. A comparison among groups was performed using the Kruskal–Wallis test followed by the uncorrected Dunn’s multiple comparison test, one-way ANOVA followed by the uncorrected Fisher’s LSD multiple comparison test, the Mann–Whitney test, or unpaired Student’s *t*-test. Results were deemed significant whenever *p* < 0.05.

## 3. Results

### 3.1. Evidence for Clinical Markers in Periodontitis and Type 2 Diabetes

Clinical periodontal parameters, such as the probing depth (PD), clinical attachment loss (CAL), plaque index (PI), and bleeding on probing (BOP) were evaluated for all the subjects, using a periodontal probe (Figure 1). Results indicate a significant increase in all parameters in both PDT and T2D-PDT patients when compared with control individuals and in T2D-PDT when compared with T2D patients (Figure 1A–D). Individuals’ distribution by age, gender, HbA1c levels, body mass index (BMI), and smoking status was organized according to diagnostic groups (Table 1). As expected, HbA1c levels were significantly increased in T2D (*p* < 0.01) and T2D-PDT (*p* < 0.0001) patients; however, no significant differences in BMI, an indicator of body fat, were observed among all groups studied (Table 1). The patient’s medication until the day of blood collection is described in Appendix A and grouped according to the Anatomical Therapeutic Chemical classification system.

Together with a slight tendency for an increase in plasma resistin levels in T2D patients (*p* = 0.092), a significant increase was observed in the T2D-PDT group, when compared with controls (*p* < 0.05) (Figure 1E). Nevertheless, a positive strong correlation was observed between plasma resistin levels and BMI (*r* = 0.6690; *p* < 0.0001) (Figure 1F), but not with CAL, BOP, or HbA1c, when data from all experimental groups was considered (data not shown).

### 3.2. Decreased Mitochondrial Oxygen Consumption in PBMCs from Periodontitis Patients

Considering the relevance of defining redox changes related to mitochondrial function and bioenergetics in PDT and T2D patients, mitochondrial respiratory activity and oxidative stress were evaluated in PBMCs isolated from all experimental groups through OCR analysis (Figure 2). Results presented in Figure 2A show average OCR traces for each group studied. Despite a slight, but non-significant, decrease (*p* < 0.05) in basal respiration (Figure 2B), a significant decrease was observed for maximal respiration (*p* < 0.05; Figure 2C), ATP production (*p* < 0.05; Figure 2D) and reserve capacity (*p* < 0.01; Figure 2E) in PDT patients compared to control individuals. All OCR parameters observed in PBMCs from T2D and T2D-PDT patients were unaltered when compared to control individuals and no significant differences were observed for H^+^ leak, non-mitochondrial respiration, and coupling efficiency when comparing all groups (Figure 2F–H).

The BHI, an alias for “mitochondrial health” [22], was shown to be decreased in PDT patients when compared to control (*p* < 0.01) and T2D-PDT (*p* < 0.05) groups (Figure 2I). These data suggest a major impact of mitochondrial dysfunction in the peripheral cells of PDT patients.

### 3.3. Enhanced H_2_O_2_ Levels and Concomitant Reduced Catalase and Glutathione Cycle in PBMCs from Periodontitis and Periodontitis plus Type 2 Diabetes

Considering the decrease found in mitochondrial respiration in PBMCs from PDT patients (Figure 2), H_2_O_2_ levels in PBMCs isolated from all groups (Figure 3A) were evaluated. We found a robust increase in H_2_O_2_ levels in PBMCs obtained from PDT patients (*p* < 0.05) which may be accounted for by reduced mitochondrial function in these cells. A significant increase in H_2_O_2_ levels was also observed in T2D-PDT patients (*p* < 0.05). No differences were observed between T2D and control subjects (Figure 3A). Since mitochondrial SOD2 (Mn-SOD) catalyzes the dismutation of the superoxide radical (O_2_•^−^) to H_2_O_2_, we investigated whether increased acetylation of SOD2 at lysine 68 (K68), which is linked to decreased enzymatic activity [24], contributed to increased ROS in PBMCs. Our results demonstrate that the ratio of acetyl(K68)-SOD2 to total SOD2 levels was unchanged and, thus, we conclude for the unaltered SOD2 activity in all patient groups (Figure 3B).

Because increased H_2_O_2_ may result from the reduced activity and/or expression levels of catalase or GPx, we further quantified these two antioxidant enzymes that reduce H_2_O_2_ into water [25]. Indeed, catalase levels were shown to be robustly and significantly decreased for PDT (*p* < 0.001), T2D (*p* < 0.01), and T2D-PDT (*p* < 0.0001) groups when compared to the control (Figure 3C), suggesting reduced catalase antioxidant defense in all patients. We further assessed changes in glutathione redox cycle enzyme parameters, namely GPx- and GRed-specific activities and the availability of GSH and GSSG, as well as the reduced and oxidized forms of glutathione, respectively. Our data showed a significant decrease by about 40% (*p* < 0.01) and 48% (*p* < 0.0001) in GPx and GRed specific activities, respectively, in T2D-PDT patients compared to control individuals (Figure 3D,F). PBMCs from T2D-PDT patients exhibited a significant decrease in GPx and GRed activities when compared with PDT (*p* < 0.05 and *p* < 0.01, respectively) or when compared with T2D patients (*p* = 0.0775 and *p* < 0.05, respectively), as depicted in Figure 3D,F, suggesting an exacerbation effect when the two diseases are present. These results are in accordance with reduced GPx-1 (*p* < 0.05) and GRed (*p* < 0.05) protein levels in T2D-PDT patients (Figure 3E,G). GSH levels were significantly decreased (*p* < 0.05) in both PDT and T2D-PDT patients, as well as GSSG levels in PDT (*p* < 0.05), T2D (*p* < 0.05) and T2D-PDT (*p* < 0.01) PBMCs in comparison to controls (Figure 3H,I). Our data highly suggest a major compromised glutathione cycle when comorbidities are present.

### 3.4. Apparent Compensatory Nrf2 and Glutamate Cysteine Ligase Protein Levels in PBMCs from Periodontitis Plus Type 2 Diabetes Mellitus

Since GCL catalyzes the first step in GSH synthesis, thus also controlling GSH levels, we analyzed both GCL relative activity and GCLc protein levels. Our results showed increased GCL activity (Figure 3J) and GCLc protein levels (Figure 3K) in T2D-PDT PBMCs compared to controls (*p* < 0.05), suggesting a compensatory mechanism to restore GSH levels when comorbidity is present. GCLc protein levels also increased (*p* < 0.01) in PBMCs from T2D patients (Figure 3K).

Nrf2 is a transcription factor controlling the expression of genes involved in cell protection against oxidative stress, including GCL [26]. Interestingly, we observed significantly increased Nrf2 levels in both T2D and T2D-PDT compared to controls (*p* < 0.05; Figure 3L). These data implicate conditions of enhanced H_2_O_2_ levels linked to accumulated Nrf2 in PBMCs, with the latter also concordant with enhanced protein levels of GCLc, one of the Nrf2 targets. This interrelationship can be better appreciated when assessing a very strong positive correlation between GCLc and Nrf2 protein levels (*r* = 0.9274, *p* <0.0001; Figure 3M, considering all groups), which suggest a potential Nrf2-dependent compensatory antioxidant mechanism.

### 3.5. Expression of Nrf2 Target Genes SOD1, GCLc, GCLm, GST, and HO-1

The levels of SOD1, GCLc, GCLm, GST, and HO-1 mRNA, Nrf2 target genes, were evaluated by RT-qPCR (Figure 4). Our data show a significant reduction in SOD1 (*p* < 0.05), GST (*p* < 0.05), GCLc (*p* < 0.001), and GCLm (*p* < 0.01) mRNA levels, as well as a tendency (*p* = 0.0972) for decreased expression of HO-1 in PBMCs from PDT patients, as compared to control (Figure 4A). Moreover, we observed a tendency for an increase in GST (*p =* 0.0792) and a significant increase in HO-1 (*p* < 0.05), GCLc (*p* < 0.01) and GCLm (*p* < 0.05) mRNA levels in PBMCs from T2D-PDT patients in comparison to PDT (Figure 4A). Moreover, the expressions of Nrf2 target genes evaluated in this study were not significantly altered in the PBMCs of T2D patients when compared to the control or T2D-PDT. We further investigated whether alterations in mRNA of Nrf2 target genes could be observed in PBMCs subpopulations. mRNA levels of Nrf2 target genes were not detected in CD3+CD4+ lymphocytes, non-CD3+ lymphocytes nor monocytes, which could be accounted for the reduced expressions of these genes in these cell populations (data not shown). Data showed that SOD1 and GST mRNA levels were reduced in CD3+CD8+ lymphocytes in both PDT (*p* < 0.0 and *p* < 0.01, respectively) and T2D-PDT patients (*p* < 0.01 and *p* < 0.001, respectively) in comparison with control individuals (Figure 4B). Unchanged HO-1 mRNA levels were observed in CD3+CD8+ lymphocytes for all groups. As observed for PBMCs, CD3+CD8+ lymphocytes from T2D patients did not reveal a significant difference in SOD-1, GST, or HO-1 mRNA levels.

## 4. Discussion

In this study, we describe changes in the mitochondrial function and oxidative status in human peripheral cells, namely PBMCs, from patients with PDT and/or T2D, two conditions that present a highly complex interrelationship. Lymphocytes, the main population present in isolated PBMCs, are a heterogeneous cell population that largely depends on mitochondria to meet its energetic demands [15,22].

We confirmed PDT patients and those with the two conditions (T2D-PDT) displayed expected periodontitis markers, namely PD, CAL, BOP, and PI, while T2D and T2D-PDT individuals showed augmented levels of HbA1c.

Periodontitis is characterized by an increase of local immune-inflammatory factors, which in turn regulate resistin mRNA levels in human PBMCs, thus indicating resistin as a possible link in the well-known association between inflammation and insulin resistance [27]. Indeed, our results showed increased levels of plasma resistin in T2D-PDT patients and a positive correlation between resistin levels and BMI when considering all data, suggesting that increased resistin levels observed in T2D-PDT may be associated with obesity and/or metabolic syndrome, as described previously [28].

Additionally, PBMCs mitochondria derived from PDT patients evidenced decreased basal and maximal respiration levels along with decreased ATP production, reserve capacity (defined as the capacity of the cell to respond to an energetic demand) and BHI values, a parameter that might reflect a potential biomarker to study pathologies influenced by oxidative stress [22]. This is particularly evident in the case of PDT patients, reinforcing compromised mitochondrial activity in PDT-derived PBMCs and reflecting mitochondrial impairment possibly due to metabolic or oxidative stress associated with periodontitis. As such, we hypothesize that mitochondrial dysfunction and related decrease in available ATP and redox deregulation impact the balance of pro- and anti-inflammatory cytokines in PDT patients. Conversely, PBMCs derived from T2D or T2D-PDT patients did not exhibit significant changes in mitochondrial OCR; this observation may be explained by the fact that all diabetic patients were under treatment with antidiabetic medication, besides antihypertensive drugs (acting at the cardiovascular system), which were prescribed to 100% of T2D, and 70% of T2D-PDT patients.

Recent studies suggested that excessive ROS production plays a role in diabetic and periodontal complications, contributing to impairment of the antioxidant gene expression responsible for ROS degradation and maintenance of vascular health. In agreement with this view, O_2_•^−^ and H_2_O_2_ can promote osteoclast formation, facilitating bone resorption and the progression of PDT [29]. Moreover, the destruction of periodontal tissues related to excessive ROS was shown to be more severe when PDT was associated with T2D [30]. Our results indicated that H_2_O_2_ production was significantly increased in PBMCs derived from PDT and T2D-PDT, but not from T2D patients. Opposing data were found showing increased ROS production and higher uncoupled oxygen consumption in PBMCs derived from diabetic patients [15]. The apparent discrepancy may be due to the fact that, in that study, only 65% of diabetic patients were under antidiabetic and antihypertensive therapy [15], whereas 100% of T2D patients in the present work are controlled for diabetes, hypertension, and hypercholesterolemia, the corresponding medication having been previously shown to reduce ROS levels [26,31,32].

Our data show reduced catalase protein levels and unchanged SOD2 activity in all patient groups. Accordingly, catalase levels and SOD1 activity were shown to be significantly decreased in red blood cells obtained from PDT patients with or without T2D, suggesting the involvement of a common factor in tissue damage [30]. In contrast, increased SOD1 and catalase activities were found in red blood cells from patients with T2D and PDT, suggesting that increased ROS production in PDT may enhance the antioxidant defense system, counterbalancing the pro-oxidant environment [33].

Our study also shows reduced GPx activity in the T2D-PDT patient’s PBMCs, evidencing decreased capacity to reduce H_2_O_2_ into H_2_O and thus decreased conversion of GSH into GSSG, along with reduced GRed activity in these patients, further diminishing the regeneration of GSH. Decreased GPx and GRed activities and higher GSSG/GSH ratio were found in saliva from diabetic patients with poor metabolic control, compared with diabetic patients with good metabolic control levels, resulting in poorer periodontal health [34]. In addition, decreased GSH levels were previously found in the periodontal tissues of patients with T2D [35]. Our results evidence that decreased levels of GSH and GSSG may be accounted for by reduced activities of GPx and GRed in PBMCs from T2D-PDT patients, contributing to enhanced ROS levels; conversely, unaltered ROS are related to a rise in GSH levels in PBMCs from T2D patients. Another study performed in the gingival tissue of PDT patients found an overexpression in peroxiredoxin (Prdx) 1 and GPx-1, whilst Prdx2 and SOD2 were up-regulated especially in the poorly controlled diabetic group with PDT; however, catalase and SOD1 expression were not significantly influenced by any of these inflammatory disorders [36]. These apparent discrepancies may be due to the differences in the biological fluid/tissue (PBMCs, red blood cells, gingival tissue).

Interestingly, our data support a compensatory role for Nrf2, a major regulator of cellular antioxidant defense, which might correlate with enhanced levels and activity of GCL (and of its catalytic subunit, GCLc), an enzyme involved in GSH synthesis, in peripheral cells from both T2D and T2D-PDT patients. We hypothesize that this can be related to regular anti-diabetic medication (e.g., metformin), with the relevant balance of antioxidant levels required to decrease the impact of increased ROS. Indeed, Nrf2 and downstream antioxidant detoxifying enzymes were shown to be activated in SH-SY5Y cells by the antidiabetic drug metformin, resulting in the inhibition of oxidative stress and mitochondrial fragmentation [26] and bone regeneration in periodontitis [37].

Our data reveal novel systemic cytopathological features of PDT and T2D-PDT linked to mitochondrial and oxidative deregulation, which may be relevant as therapeutical targets.

Data also evidence a significant reduction in SOD1, GST, GCLc and GCLm mRNA levels, while a tendency for decreased expression of HO-1 in PBMCs from PDT patients, in accordance with findings obtained by Sima and co-authors [38] showing decreased levels of SOD1 and catalase in PDT patients oral polymorphonuclear leukocytes, when compared to controls. Enhanced local and systemic oxidative damage and Nrf2 downregulation (contrasting with our Nrf2 data), along with decreased SOD activity were shown in rat periodontal tissues, which were aggravated by diabetic conditions [17]. Enhanced GST and HO-1 mRNA levels in T2D-PDT, compared to PDT patients, might be explained by an attempt to afford extra biological protection against superoxide generation observed in hyperglycemic states [35]. Our results suggest that increased Nrf2 levels in PBMCs from T2D-PDT patients are not effective in activating the Nrf2 pathway and the transcription of all its target genes.

Lymphocytes are heterogeneous cell populations in which CD3+CD4+ and CD3+CD8+ subsets appear to assume important functions under bacterial infections that occur in PDT. In fact, in gingivitis conditions, the CD3+CD4+/CD3+CD8+ ratio approximately accounted for 2:1 [39], whereas gingival cells from PDT lesions revealed a 1:1 CD3+CD4+/CD3+CD8+ ratio [40]. These observations demonstrated an important increase of CD3+CD8+ T cells with pathology progression. Considering the importance of CD3+CD8+ lymphocytes in PDT’s inflammatory pathways, this might underlie the detected expression of Nrf2 target genes in this PBMCs’ subpopulation. Interestingly, the CD3+CD8+ lymphocytes subpopulation presented similar results to those observed in PBMCs with respect to reduced SOD1 and GST mRNA levels in PDT; however, decreased levels of SOD1 and GST mRNA levels were also observed in T2D-PDT patients CD3+CD8+ lymphocytes, in comparison to control individuals. Although anti-diabetic medication could play a role in ameliorating the Nrf2 pathway in PBMCs, it does not seem to influence the subpopulation of CD3+CD8+ lymphocytes.

## 5. Conclusions

The present study clearly shows reduced mitochondrial activity accompanied by increased ROS, reduced catalase levels, and decreased mRNA levels of candidate Nrf2 target genes in PBMCs, including CD3+CD8+ lymphocytes, from less pharmacologically treated PDT patients. Of relevance, PBMCs from T2D-PDT exhibit a decrease in catalase and overall glutathione redox cycle activity concomitant with enhanced ROS levels despite anti-diabetic treatment. In addition, enhanced Nrf2-dependent targets, such as GCLc, GCLm, GST, or HO-1, may act to counterbalance the cellular antioxidant profile in PBMCs, but not CD3+CD8+ cells, from T2D-PDT patients. Determining the mechanisms that lead to increased oxidative stress in patients with PDT and T2D is expected to contribute to identifying new therapeutic targets to better control the progression of PDT in these patients and, thus, improve the metabolic control of patients with T2D-PDT comorbidity.

## Figures and Tables

**Figure 1 antioxidants-12-00226-f001:**
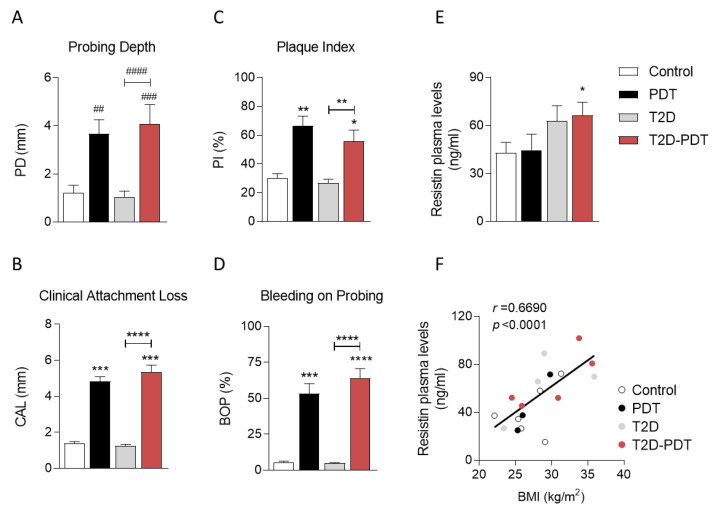
Clinical parameters for periodontitis evaluation, blood plasma resistin levels, and correlation between plasma resistin levels and body mass index (BMI). Values of probing depth (**A**), clinical attachment loss (**B**), plaque index (**C**), bleeding on probing (**D**), and (**E**) resistin levels evaluated in plasma obtained from control individuals, PDT, T2D, and T2D-PDT patients. (**F**) correlation of resistin levels with BMI values of the same patients for all groups. Data are presented as the mean ± SEM from 3–10 individuals per group. Statistical analysis: ^##^
*p* < 0.01; ^###^
*p* < 0.001; ^####^
*p* < 0.001 by the Kruskal–Wallis test with multiple comparisons; * *p* < 0.05; ** *p* < 0.01; *** *p* < 0.001; **** *p* < 0.001 by one-way ANOVA with multiple comparisons, when compared to the respective control.

**Figure 2 antioxidants-12-00226-f002:**
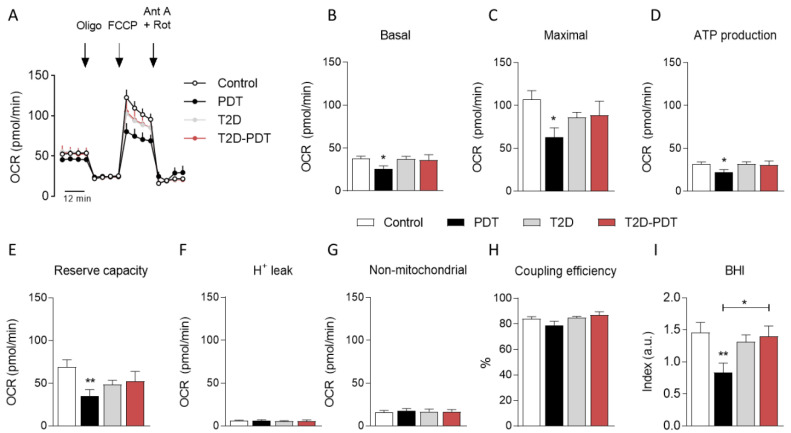
Oxygen consumption rate (OCR) in PBMCs. (**A**) Average OCR traces before and following oligomycin (Oligo), carbonyl cyanide 4-(trifluoromethoxy)phenylhydrazone (FCCP), and antimycin plus rotenone injection (Ant A + Rot); (**B**) basal respiration; (**C**) maximal respiration; (**D**) ATP production; (**E**) reserve capacity; (**F**) H^+^ leak; (**G**) non-mitochondrial respiration; (**H**) coupling efficiency; (**I**) bioenergetic health index (BHI) calculated accordingly to the formula: BHI = log [(reserve capacity × ATP-linked)/non-mitochondria respiration × proton leak)]. OCR results are presented as the mean ± SEM of independent experiments expressed in nmoles O_2_/mL/min/0.2 × 10^6^ cells performed in PBMCs isolated from control individuals, PDT, T2D, and T2D-PDT patients, 10 individuals per group, run in duplicates or triplicates. Statistical analysis: * *p* < 0.05 and ** *p* < 0.01 by the Kruskal–Wallis test with multiple comparisons, when compared to the respective control.

**Figure 3 antioxidants-12-00226-f003:**
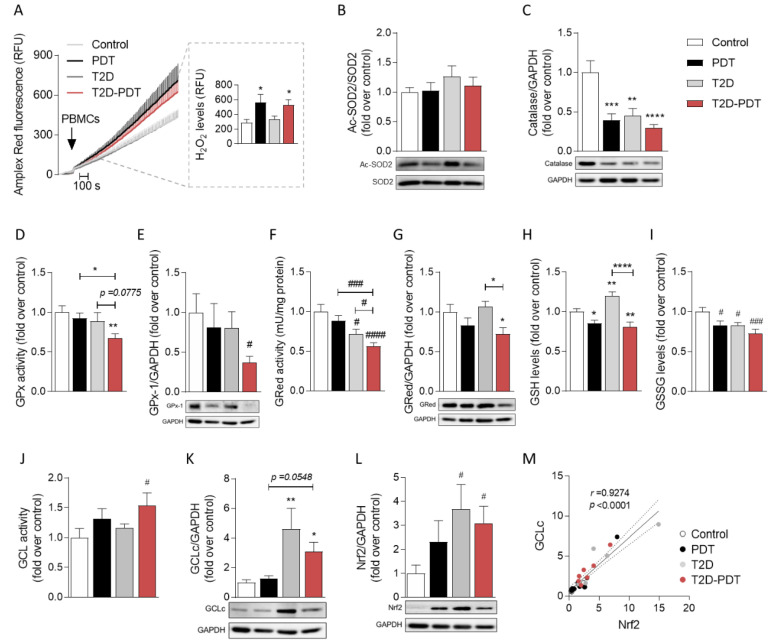
H_2_O_2_ production, antioxidant profile, and Nrf2 levels in PBMCs. (**A**) Representative traces of H_2_O_2_ production evaluated using the Amplex Red assay and quantification of H_2_O_2_ production (slope/15 min; inset); (**B**) Ac-SOD2/SOD2 (24/25 kDa) and (**C**) catalase (60 kDa) protein levels; (**D**) GPx activity and (**E**) GPx-1 (22 kDa) protein levels, (**F**) GRed activity and (**G**) protein levels (58 kDa); (**H**) reduced glutathione (GSH) levels and (**I**) oxidized glutathione (GSSG) levels; (**J**) GCL activity and (**K**) GCLc (73 kDa) protein levels; (**L**) Nrf2 (68 kDa) protein levels and (**M**) correlation between GCLc and Nrf2 protein levels. GAPDH (36 kDa) was used as control loading for western blots. Data are presented as the mean ± SEM, performed in PBMCs isolated from control individuals, PDT, T2D, and T2D-PDT patients, 10 individuals *per* group, run in duplicates to quadruplicates. Statistical analysis: ^#^
*p* < 0.05; ^###^
*p* < 0.001; ^####^
*p* < 0.001 by the Kruskal–Wallis test with multiple comparisons; * *p* < 0.05; ** *p* < 0.01; *** *p* < 0.001; **** *p* < 0.001 by one-way ANOVA with multiple comparisons, when compared to the respective control.

**Figure 4 antioxidants-12-00226-f004:**
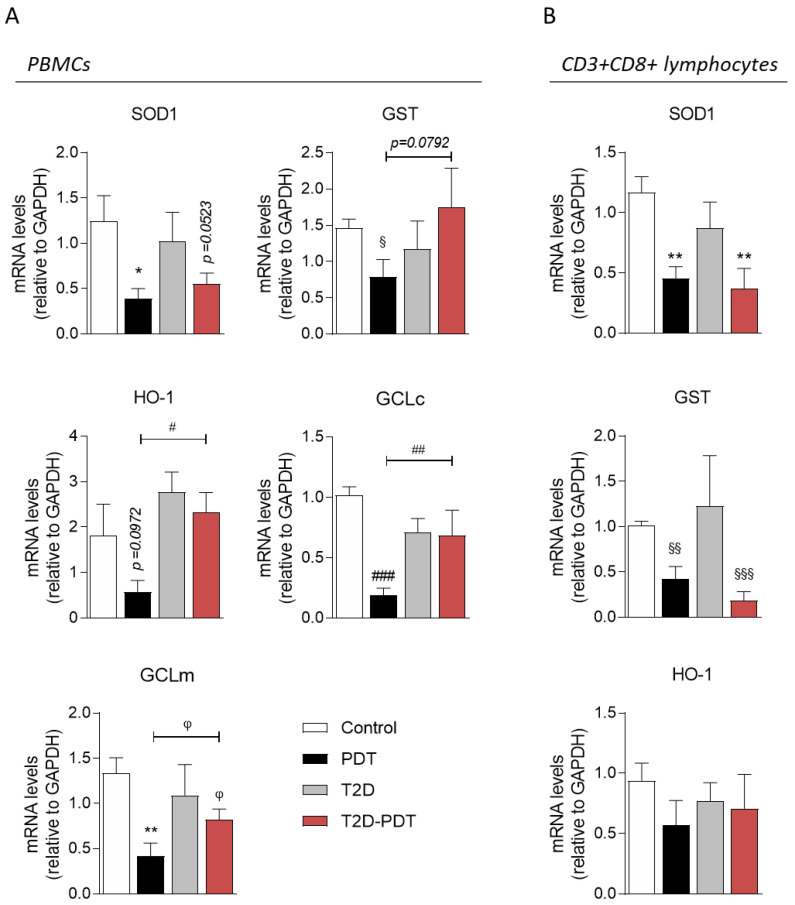
mRNA levels of Nrf2 target genes in PBMCs and CD3+CD8+ lymphocytes. Relative expression of (**A**) SOD1, GST, HO-1, GCLc, and GCLm mRNA in PBMCs or (**B**) SOD1, GST, and HO-1 in CD3+CD8+ lymphocytes, from control individuals, PDT, T2D, and T2D-PDT patients PBMCs prepared as described in the Section 2. GAPDH was used as the housekeeping messenger. Data are presented as the mean ± SEM from 4 individuals per group. Statistical analysis: * *p* < 0.05; ** *p* < 0.01 by one-way ANOVA with multiple comparisons; ^#^
*p* < 0.05; ^##^
*p* < 0.01; ^###^
*p* < 0.001 by the Kruskal–Wallis test with multiple comparisons; ^§^
*p* < 0.05; ^§§^
*p* < 0.01; ^§§§^
*p* < 0.001 by unpaired Student’s *t*-test; ^φ^
*p* < 0.05 by the Mann–Whitney test when compared to the respective control.

**Table 1 antioxidants-12-00226-t001:** Subject characterization according to age, gender, and clinical parameters.

Subject	Age	Males/Females	HbA1c	BMI	Smokers
	(Years)	(Number)	(%)	(kg/m^2^)	(Number)
Control	59.5 ± 1.7	1/9	5.74 ± 0.06	26.85 ± 0.96	0
PDT	56.0 ± 2.1	7/3	5.72 ± 0.12	27.20 ± 1.11	3
T2D	49.6 ± 10.4	4/6	6.41 ± 0.19 ^##^	28.11 ± 1.51	0
T2D-PDT	53.6 ± 7.7	8/2	7.08 ± 0.22 ^####^	29.18 ± 1.40	1

Subjects were categorized per age, gender, glycated hemoglobin (HbA1c), body mass index (BMI), and smoking status, and classified in accordance with clinical evaluation for periodontitis (PDT), type 2 diabetes (T2D) and type 2 diabetes plus periodontitis (T2D-PDT). Data are presented as the mean ± SEM from 10 individuals per group. Statistical analysis: ^##^
*p* < 0.01; ^####^
*p* < 0.0001 by the Kruskal–Wallis test with multiple comparisons when compared to the control.

## Data Availability

Data presented in this study are available on request from the corresponding authors. The data are not publicly available due to ethical restrictions.

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
