# Peer review of "Mitochondrial and Redox Changes in Periodontitis and Type 2 Diabetes Human Blood Mononuclear Cells"

_antioxidants, 2023, doi:10.3390/antiox12020226_

Round 1
Reviewer 1 Report
The aim of the study was to assess mitochondrial changes and redox regulation in PBMCs from patients with periodontitis and/or type 2 diabetes.
The study is very clear and the methodology seems strong. I particularly appreciated the care taken in the statistical analysis and the calculation of the number of subjects needed.
I have consequently little to criticize.
First of all, I disagree with an assertion made by the authors in the introduction. They say "Periodontitis (PDT) is a chronic oral inflammatory disease, initiated by the accumulation of a pathogenic subgingival biofilm, and within which microbial dysbiosis leads to a chronic dysregulated immune-inflammatory response".
I know that cause of peridontitis is still in debate but several studies point out the initiation of the disease to be rather linked to an inappropriate host immune response (see papers of Bartold et al.). Thus I would reconsider this sentence, better saying "Periodontitis (PDT) is a chronic oral inflammatory disease, linked to an inappropriate host immune response to a dysbiotic microbiome".
In M&M, the authors defined periodontitis as " interdental CAL was detectable at ≥ 2 non‐adjacent teeth, probing depth over 3 mm and the proportion of sites that bleed on probing". Please add the considered proportion.
In M&M the authors spoke about "uncontrolled systemic diseases (except T2D) or neoplasms". Did any subjects have such pathologies? Which ones?
The differences between the 3 groups of T2D, T2D+PDT and PDT end up being confusing. Especially with all oxidant/antioxidant systems. It could be considered to make a figure to summarize the different actions demonstrated.
Finally, I would miss some discussion of the role of findings in explaining the cause and/or phenotype observed in periodontitis. Does the metabolic profile of PBMC correlate with the clinical parameters of periodontitis in this study and can biological hypotheses be formulated?
Author Response
The aim of the study was to assess mitochondrial changes and redox regulation in PBMCs from patients with periodontitis and/or type 2 diabetes.
The study is very clear and the methodology seems strong. I particularly appreciated the care taken in the statistical analysis and the calculation of the number of subjects needed.
I have consequently little to criticize.
First of all, I disagree with an assertion made by the authors in the introduction. They say "Periodontitis (PDT) is a chronic oral inflammatory disease, initiated by the accumulation of a pathogenic subgingival biofilm, and within which microbial dysbiosis leads to a chronic dysregulated immune-inflammatory response".
I know that cause of peridontitis is still in debate but several studies point out the initiation of the disease to be rather linked to an inappropriate host immune response (see papers of Bartold et al.). Thus I would reconsider this sentence, better saying "Periodontitis (PDT) is a chronic oral inflammatory disease, linked to an inappropriate host immune response to a dysbiotic microbiome".
R: We thank the reviewer for the correction. The sentence was rephrased and the new reference included in the list of references as [1].
In M&M, the authors defined periodontitis as " interdental CAL was detectable at ≥ 2 non‐adjacent teeth, probing depth over 3 mm and the proportion of sites that bleed on probing". Please add the considered proportion.
R: We thank the reviewer for the comment. Actually, we did not establish a threshold for a proportion of sites for bleeding on probing, but registered this important clinical parameter of periodontal inflammation. The sentence is corrected to “(…) interdental CAL was detectable at ≥ 2 non‐adjacent teeth, probing depth over 3 mm and bleeding on probing” (page 6).
In M&M the authors spoke about "uncontrolled systemic diseases (except T2D) or neoplasms". Did any subjects have such pathologies? Which ones?
R: We thank the reviewer for the comment. Such pathologies were referred as exclusion criteria. So, no patients included had those conditions, such as rheumatoid arthritis, cardiac diseases, Parkinson’s disease.
The differences between the 3 groups of T2D, T2D+PDT and PDT end up being confusing. Especially with all oxidant/antioxidant systems. It could be considered to make a figure to summarize the different actions demonstrated.
R: The graphical abstract aims to summarize the most relevant results concerning PDT and T2D-PDT conditions, the two main focus of the present work.
Finally, I would miss some discussion of the role of findings in explaining the cause and/or phenotype observed in periodontitis. Does the metabolic profile of PBMC correlate with the clinical parameters of periodontitis in this study and can biological hypotheses be formulated?
R: Compromised mitochondrial activity and related oxidative stress is described in Discussion (page 15); here we anticipate that “mitochondrial impairment possibility due to metabolic or oxidative stress associated with periodontitis” and include the hypothesis that “(…) mitochondrial dysfunction and related decrease in available ATP and redox deregulation impact on the balance of pro- and anti-inflammatory cytokines in PDT patients”.

Reviewer 2 Report
This is an interesting study conducted on PBMCs, as well as on lymphocytes, obtained from patients with diabetes, with or without the presence of periodontitis comorbidity, in order to detect any changes in mitochondrial bioenergetics and redox regulation. The experimental design is clearly outlined, methodology accurately described with a proper power analysis for the study design, results presented in a logical and clear sequence, and discussion well-constructed to consider the data in the context of knowledge of the patho-physiology. The study is very convincing and adds new information useful to understand the periodontitis complication occurring in patients affected by diabetes, in order to suggest a more efficacious prevention and/or therapy.
I may suggest just some minor points to be considered.
In Figure 1F and 3M, the coefficient of determination r2 is reported. Since the aim is not to indicate how successful the regression was in explaining the response, but instead the strength of the linear relationship between the two variables, then the Coefficient of Correlation (Pearson’s r, that is the square root of r2) could be more appropriate.
Figure 2B: please indicate that the value shown over the PDT bar is significance; p=0.0614. Same comment for Figure 4A, where several p values are indicated in the graphs, but without the mention that the values are “p”.
Figure 4A,B: please uniform the symbols used for significance; “t”, “tt”” and “ttt” for GST/GCLc panels should be replaced, since no mention to these symbols has been provided in figure legend.
Figure 4 legend: “Statistical analysis: *p<0.05; **p<0.01 by one-way ANOVA followed by Kruskal-Wallis test with multiple comparisons or unpaired Student’s t-test.”: It is not clear the statistical analysis here performed: the Kruskal–Wallis test is one-way ANOVA on ranks, that is a non-parametric analogue of ANOVA; therefore it is not appropriate, as stated in the legend, to perform one-way ANOVA “followed” by Kruskal-Wallis test, since Kruskal-Wallis test is alternative to ANOVA, not a post-hoc test. At page 7, under “Statistical analysis” paragraph, it is stated “Comparison among groups was performed via nonparametric Kruskal-Wallis test without multiple comparisons correction (uncorrected Dunn’s test).”, without any mention to ANOVA and/or Student’s t test. Moreover, it is not clear regarding the asterisks of Figure 4, to which comparisons they refer (vs control?). Similar comment also for legend of figure 2, when referring to “Statistical analysis: *p <0.05 and **p <0.01 by Kruskal-Wallis test.”, and of figure 3.
Page 7, Statistical analysis: “In order to determine a significant outlier, Grubbs’ test was performed.” It is not stated what happened if a significant outlier (if any found) occurred.
Page 12 lines 36-38: “Moreover, we observed a tendency for an increase in HO1 (p =0.072) and a significant increase in GST (p <0.05) mRNA levels in PBMCs from T2D-PDT patients in comparison to PDT (Figure 4A).”: Comparing with Figure 4A, the significance values for HO-1 and GST mRNA levels seem inverted.
Throughout the manuscript: please uniform the italic style for “p”
Author Response
This is an interesting study conducted on PBMCs, as well as on lymphocytes, obtained from patients with diabetes, with or without the presence of periodontitis comorbidity, in order to detect any changes in mitochondrial bioenergetics and redox regulation. The experimental design is clearly outlined, methodology accurately described with a proper power analysis for the study design, results presented in a logical and clear sequence, and discussion well-constructed to consider the data in the context of knowledge of the patho-physiology. The study is very convincing and adds new information useful to understand the periodontitis complication occurring in patients affected by diabetes, in order to suggest a more efficacious prevention and/or therapy.
I may suggest just some minor points to be considered.
In Figure 1F and 3M, the coefficient of determination r2 is reported. Since the aim is not to indicate how successful the regression was in explaining the response, but instead the strength of the linear relationship between the two variables, then the Coefficient of Correlation (Pearson’s r, that is the square root of r2) could be more appropriate.
R: We thank the reviewer for the suggestion. The Pearson correlation coefficients (Pearson’s r) were calculated and now presented in Figures 1F and 3M, and described in pages 12 and 14.
Figure 2B: please indicate that the value shown over the PDT bar is significance; p=0.0614. Same comment for Figure 4A, where several p values are indicated in the graphs, but without the mention that the values are “p”.
R: We thank the reviewer for the comment. The “p” symbol is now included in the p values in both Figures 2B and 4A.
Figure 4A,B: please uniform the symbols used for significance; “t”, “tt”” and “ttt” for GST/GCLc panels should be replaced, since no mention to these symbols has been provided in figure legend.
R: The symbols were changed and are now mentioned in the legend of Figure 4 (§p <0.05; §§p <0.01; §§§p <0.001).
Figure 4 legend: “Statistical analysis: *p<0.05; **p<0.01 by one-way ANOVA followed by Kruskal-Wallis test with multiple comparisons or unpaired Student’s t-test.”: It is not clear the statistical analysis here performed: the Kruskal–Wallis test is one-way ANOVA on ranks, that is a non-parametric analogue of ANOVA; therefore it is not appropriate, as stated in the legend, to perform one-way ANOVA “followed” by c test, since Kruskal-Wallis test is alternative to ANOVA, not a post-hoc test.
R: We thank the reviewer for the comment. Indeed, the previous description was not correct. It is now stated as: “Statistical analysis: (…) by one-way ANOVA with multiple comparisons; (…)by Kruskal-Wallis test with multiple comparisons; (…).”.
At page 7, under “Statistical analysis” paragraph, it is stated “Comparison among groups was performed via nonparametric Kruskal-Wallis test without multiple comparisons correction (uncorrected Dunn’s test).”, without any mention to ANOVA and/or Student’s t test. Moreover, it is not clear regarding the asterisks of Figure 4, to which comparisons they refer (vs control?). Similar comment also for legend of figure 2, when referring to “Statistical analysis: *p <0.05 and **p <0.01 by Kruskal-Wallis test.”, and of figure 3.
R: The “Statistical analysis” paragraph is now corrected and the comparison to the respective control is now described in the figure legends.
Page 7, Statistical analysis: “In order to determine a significant outlier, Grubbs’ test was performed.” It is not stated what happened if a significant outlier (if any found) occurred.
R: The sentence was added by mistake and was now removed from the statistical analysis paragraph.
Page 12 lines 36-38: “Moreover, we observed a tendency for an increase in HO1 (p =0.072) and a significant increase in GST (p <0.05) mRNA levels in PBMCs from T2D-PDT patients in comparison to PDT (Figure 4A).”: Comparing with Figure 4A, the significance values for HO-1 and GST mRNA levels seem inverted.
R: We thank the reviewer for pointing out this error. The description of significant values is now in accordance with the respective mRNA levels for GST and HO-1 (page 14).
Throughout the manuscript: please uniform the italic style for “p”
R: The italic style for “p” was uniformized.

Reviewer 3 Report
The article of Ferreira I.L. et al elucidates an important problem of the relationship between periodontitis (PDI) and type 2 diabetes (T2D), two common chronic diseases in humans. 4 groups of 10 patients each were analyzed. The authors claimed that all of them were balanced by body weight, age and gender characteristics. At the same time a quick glance on the table 1 demonstrates that control group is dominated by women (9 females) while experimental groups are staffed predominantly by men. There are also doubts about age balance. Formally in the table all groups are very well balanced by age but age deviation 0 in control group is surprising. The authors mention in the text the range in age from 45 to 75 years. At the same time taking into account age deviations there is no one in the table aged 75 years which is doubtful. Age deviations in the table must be corrected.
The graphical presentation of the data should be also improved. On Fig.2A the data for T2D patients are not seen may be because of color selection. On Fig.3A illustrating fluorescent response of Amplex Red control data is absent. This graphical failures must be corrected.
There are certain complaints about the Discussion section. First, the statement “diabetes might be associatedwith higher uncoupled oxygen consumption and thus higher ROS production” is absolutely wrong. ROS production is associated with high membrane potential and mild uncoupling is one of the effective approaches to reduce ROS production. Second, the data obtained pointed that in some cases, for BHI factor for instance, T2D positively affected the state of the peripheral human blood mononuclear cells (PBMCs), the main experimental model of the authors. This interesting and important result should be discussed in more details may be with individual analysis of the possible action of antidiabetic drags.
These remarks do not detract from the undoubted significance of the work, which can be accepted for publication after making the appropriate corrections.
Author Response
The article of Ferreira I.L. et al elucidates an important problem of the relationship between periodontitis (PDI) and type 2 diabetes (T2D), two common chronic diseases in humans.
4 groups of 10 patients each were analyzed. The authors claimed that all of them were balanced by body weight, age and gender characteristics. At the same time a quick glance on the table 1 demonstrates that control group is dominated by women (9 females) while experimental groups are staffed predominantly by men. There are also doubts about age balance. Formally in the table all groups are very well balanced by age but age deviation 0 in control group is surprising. The authors mention in the text the range in age from 45 to 75 years. At the same time taking into account age deviations there is no one in the table aged 75 years which is doubtful. Age deviations in the table must be corrected.
R: We thank the reviewer for raising these questions. Although we aimed to recruit a similar number of women and men, we were not able to do it. Indeed, we have more women than men in controls and more men in PDT and T2D-PDT disease groups; based on these observations we eliminated part of the sentence in ‘Study Design’ related with gender balance. We could only argue that gender balance is maintained in overall study when adding all women and men of all groups. There was a mistake in the deviation of age of the control group, which has been corrected (Table 1). The age range (39-62 years old) was also corrected in the ‘Study Design’.
The graphical presentation of the data should be also improved. On Fig.2A the data for T2D patients are not seen may be because of color selection.
R: We thank the reviewer for the suggestion and the graphical presentation of Fig. 2A was improved.
On Fig.3A illustrating fluorescent response of Amplex Red control data is absent. This graphical failures must be corrected.
R: We thank the reviewer for the suggestion; the graphical presentation of Fig. 3A was improved to better visualize the control Amplex Red fluorescence.
There are certain complaints about the Discussion section.
First, the statement “diabetes might be associated with higher uncoupled oxygen consumption and thus higher ROS production” is absolutely wrong. ROS production is associated with high membrane potential and mild uncoupling is one of the effective approaches to reduce ROS production.
R: We certainly agree with the reviewer that mitochondrial ROS production occurs under maintained/high mitochondrial membrane potential. The sentence refers to reference nº 15 and thus it was rephrased to avoid misinterpretation: “Opposing data was found showing increased ROS production and higher uncoupled oxygen consumption in PBMCs derived from diabetic patients [15]”.
Second, the data obtained pointed that in some cases, for BHI factor for instance, T2D positively affected the state of the peripheral human blood mononuclear cells (PBMCs), the main experimental model of the authors. This interesting and important result should be discussed in more details may be with individual analysis of the possible action of antidiabetic drags.
R: As described in the “Discussion section”, 100% of T2D patients evaluated in the present work are controlled for diabetes, hypertension and hypercholesterolemia (Table 2, supplementary data). These medications were previously shown to reduce ROS levels, as in [25] for antidiabetics, [30] for antihypertensive and [31] for statins. In addition, the increase in Nrf2 protein levels observed in T2D and T2D-PDT patients in our work can be related with regular anti-diabetic medication (e.g. metformin) prescribed in these patients, as stated in Table 2, which may balance antioxidant levels required to decrease the impact of increased ROS. Indeed, Nrf2 and downstream antioxidant detoxifying enzymes were shown to be activated in SH-SY5Y cells by the antidiabetic drug metformin, resulting in the inhibition of oxidative stress and mitochondrial fragmentation [25] and in bone regeneration in periodontitis [36], as described in the Ms.
These remarks do not detract from the undoubted significance of the work, which can be accepted for publication after making the appropriate corrections.
R: We thank the reviewer for this positive comment.

Reviewer 4 Report
The manuscript of “Mitochondrial and redox changes in periodontitis and type 2 diabetes human blood mononuclear cells” by Ildete L Ferreira and co-authors aims to study mitochondrial and redox (de)regulation in peripheral blood mononuclear cells (PBMCs) in periodontitis (PDT) patients, compared to type 2 diabetes (T2D) and comorbid T2D-PDT patients and control individuals. The authors demonstrated that reduced mitochondrial activity was accompanied by increased ROS, reduced catalase levels and decreased mRNA levels of candidate Nrf2 target genes in PBMCs, including CD3+CD8+ lymphocytes, from less pharmacologically treated PDT patients. In parallel, PBMCs from T2D-PDT patients exhibited a decrease in catalase and over-all glutathione redox cycle activity concomitant with enhanced ROS levels despite anti-diabetic treatment. The authors suggested that enhanced Nrf2-dependent targets, as GCLc, GST or HO-1, can act to counterbalance the cellular antioxidant profile in PBMCs, but not CD3+CD8+ cells, from T2D-PDT patients.
The study was conducted at a high level. The manuscript is interesting, well written and has a high degree of originality and novelty.
Comments.
1. PBMCs are known to consist of different cell types (monocytes and lymphocytes). It is also well known that the progression of T2DM is associated with disturbances of immune status that can be reflected by alterations of the profile of circulating immune cells, including monocytes and lymphocytes (doi: 10.1155/2019/1491083). Therefore, it is very important to provide data on what types of cells and in what percentage are represented in the PBMC samples. In the current study, the authors mentioned that “lymphocytes, the main population present in isolated PBMCs, are a heterogeneous cell population that largely depends on mitochondria to meet its energetic demands [15,21].” The authors should describe in more detail the percentage of different cell types in the samples of peripheral blood mononuclear cells (PBMCs) in 4 experimental groups. It would also be important to determine changes in the ratio of helper and cytotoxic T lymphocytes in the tested groups.
2. According to the Materials and Methods (The Fluorescence activated cell sorting (FACS) section), the authors performed the cell sorting procedure to separate CD3+CD4+ and CD3+CD8+ lymphocyte populations. However, no data on CD3+CD4+ lymphocytes was provided. Please, explain.
3. The authors should discuss the possible advantages of using a mixed preparation of monocytes and lymphocytes in comparison with erythrocytes or platelets, which are widely used for rapid studies of mitochondrial function in T2DM.
4. The authors described the methods of statistical analysis in different ways in the Statistical analysis section and in the legends to the figures (Kruskal-Wallis test without multiple comparisons vs. Kruskal-Wallis test with multiple comparisons or unpaired Student’s t-test). Please, check it out.
Author Response
The manuscript of “Mitochondrial and redox changes in periodontitis and type 2 diabetes human blood mononuclear cells” by Ildete L Ferreira and co-authors aims to study mitochondrial and redox (de)regulation in peripheral blood mononuclear cells (PBMCs) in periodontitis (PDT) patients, compared to type 2 diabetes (T2D) and comorbid T2D-PDT patients and control individuals. The authors demonstrated that reduced mitochondrial activity was accompanied by increased ROS, reduced catalase levels and decreased mRNA levels of candidate Nrf2 target genes in PBMCs, including CD3+CD8+ lymphocytes, from less pharmacologically treated PDT patients. In parallel, PBMCs from T2D-PDT patients exhibited a decrease in catalase and over-all glutathione redox cycle activity concomitant with enhanced ROS levels despite anti-diabetic treatment. The authors suggested that enhanced Nrf2-dependent targets, as GCLc, GST or HO-1, can act to counterbalance the cellular antioxidant profile in PBMCs, but not CD3+CD8+ cells, from T2D-PDT patients.
The study was conducted at a high level. The manuscript is interesting, well written and has a high degree of originality and novelty.
R: We thank the reviewer for this positive comment.
Comments.
- PBMCs are known to consist of different cell types (monocytes and lymphocytes). It is also well known that the progression of T2DM is associated with disturbances of immune status that can be reflected by alterations of the profile of circulating immune cells, including monocytes and lymphocytes (doi: 10.1155/2019/1491083). Therefore, it is very important to provide data on what types of cells and in what percentage are represented in the PBMC samples. In the current study, the authors mentioned that “lymphocytes, the main population present in isolated PBMCs, are a heterogeneous cell population that largely depends on mitochondria to meet its energetic demands [15,21].” The authors should describe in more detail the percentage of different cell types in the samples of peripheral blood mononuclear cells (PBMCs) in 4 experimental groups. It would also be important to determine changes in the ratio of helper and cytotoxic T lymphocytes in the tested groups.
R: In this study, PBMCs were separated in CD3+CD4+, CD3+CD8+, non CD3+ and monocytes, as now fully described in “Fluorescence Activated Cell Sorting (FACS)” section (page 9) (Figure S1.A, below) and the cell number for each population is described below. No major differences were observed in the percentage of lymphocyte subpopulations and monocytes obtained among the groups (Figure S1.B, below). Figure S1 was included in ‘Supplementary Material’ and referred in the Ms (page 9). Unfortunately, at this stage, we will not be able to determine changes in the ratio of helper and cytotoxic T lymphocytes.
Figure S1. FACS separation of PBMCs subpopulations. PBMCs derived from control individuals, PDT, T2D and PDT-T2D patients were sorted (A) into CD3+CD4+ lymphocytes, CD3+CD8+ lymphocytes, non CD3+ lymphocytes and monocytes and the number of cells counted by FACS (B). Data are presented in percentage (%) of cells evaluated as the mean ± SEM.
- According to the Materials and Methods (The Fluorescence activated cell sorting (FACS) section), the authors performed the cell sorting procedure to separate CD3+CD4+ and CD3+CD8+ lymphocyte populations. However, no data on CD3+CD4+ lymphocytes was provided. Please, explain.
R: Unfortunately, “mRNA levels of Nrf2 target genes were not detected in CD3+CD4+ lymphocytes, non CD3+ lymphocytes or monocytes, which could be accounted for reduced expression of these genes in these cell populations (data not shown)”. This information is described in “Expression of Nrf2 Target Genes SOD1, GCLc, GCLm, GST and HO-1” subsection of the Results section (page 14).
- The authors should discuss the possible advantages of using a mixed preparation of monocytes and lymphocytes in comparison with erythrocytes or platelets, which are widely used for rapid studies of mitochondrial function in T2DM.
R: Throughout the study, we used PBMCs that are a mix of monocytes and lymphocytes. Nrf2-regulated gene expression was performed in PBMCs and in mononuclear cells subpopulations, as described in the Ms. Since differentiated platelets and erythrocytes do not have nucleus, they cannot be used to study gene expression; furthermore, differentiated erythrocytes do not have mitochondria precluding the study of mitochondrial function.
- The authors described the methods of statistical analysis in different ways in the Statistical analysis section and in the legends to the figures (Kruskal-Wallis test without multiple comparisons vs. Kruskal-Wallis test with multiple comparisons or unpaired Student’s t-test). Please, check it out.
R: We thank the reviewer for the suggestion. The description of statistical analysis used was corrected throughout the manuscript.

Round 2
Reviewer 4 Report
I have no more comments.